# Fault reactivation by gas injection at an underground gas storage off the east coast of Spain

Antonio Villaseñor[1], Robert B. Herrmann[2], Beatriz Gaite[1,*] and Arantza Ugalde[1,**]

[1]Institute of Earth Sciences Jaume Almera, CSIC, 08028 Barcelona, Spain
[2]Department of Earth and Atmospheric Sciences, Saint Louis University, MO 63108, USA
[*]Now at: Instituto Geográfico Nacional, 28003 Madrid, Spain
[**]Now at: Institute of Marine Sciences, CSIC, 08003 Barcelona, Spain

*Correspondence to*: Antonio Villaseñor (antonio.villasenor@csic.es)

**Abstract.** During September-October of 2013 an intense swarm of earthquakes occurred off the east coast of Spain associated with the injection of the base gas in an offshore underground gas storage. Two weeks after the end of the injection operations, three moderate-sized earthquakes ($M_w$ 4.0-4.1) occurred near the storage. These events were widely felt by the nearby population, leading to the indefinite shut-down of the facility. Here we investigate the source parameters (focal depth and mechanism) of the largest earthquakes in the sequence in order to identify the faults reactivated by the gas injection, and to help understand the processes that caused the earthquakes. Our waveform modeling results indicate that the largest earthquakes occurred at depths of 6-8 km beneath the sea floor, significantly deeper than the injection depth (~ 1800 m). Although we cannot undoubtedly discriminate the fault plane from the two nodal planes of the mechanisms, most evidence seems to favor a NW-SE striking fault plane. We propose that the gas injection reactivated faults in the Paleozoic basement, with regional orientation possibly inherited from the opening of the Valencia Trough.

## 1 Introduction

Induced seismicity is a growing hazard, as industrial activities that involve the injection and/or extraction of fluids become more common and closer to populated areas. A recent episode of induced seismicity (September-October 2013) occurred at the CASTOR underground gas storage (UGS). The CASTOR UGS was redeveloped in the depleted Amposta oil field (Seeman et al., 1990) located 22 km off the east coast of Spain, south of the Ebro delta (Figure 1). Water depth at the location of the storage is 61 m. At the time of the earthquake sequence, the seismic monitoring network for the facility consisted only of two short period stations located inland (> 25 km distance from the UGS), and was complemented with existing stations from other regional networks (Figure 1). No ocean bottom seismometer (OBS) was located close to the platform. This poor monitoring configuration, lacking nearby stations, made it difficult to locate earthquakes accurately and particularly to constrain their focal depths. A previous study (Cesca et al., 2014) found shallow focal depths for most of the earthquakes (approximately 2 km), consistent with the injection depth of ~1.8 km. More recently Gaite et al. (2016) obtained new locations using a 3-D model developed for the study region, and refined arrival time picks through waveform cross

correlation. As a result of this analysis they obtained focal depths centered at 6 km. Saló et al. (2017) have also obtained focal mechanisms whose depths are similar to those of Gaite et al. (2016). Finally Juanes et al. (2017) found depths slightly shallower than 5 km using a 1-D flat layered model, and a range of deeper depths when using a 3-D model. This discrepancy between studies is small considering the errors associated with locations based on arrival times alone, particularly when there are no nearby stations to the earthquakes as in this case. However, the difference is significant in terms of the processes responsible for the seismicity and for the identification of the reactivated faults. Shallow focal depths could indicate that the earthquakes were induced directly by the gas injection. On the other hand, deeper focal depths would suggest that the events were triggered in more distant faults that were critically stressed, either by pore-pressure changes or other mechanisms (Ellsworth, 2013;Bhattacharya and Viesca, 2019). While deeper events represent a lower hazard for the seal of the storage, they could potentially be of larger magnitude and affect the facility and nearby population.

Therefore, in order to better constrain focal depths, we have used the sensitivity of seismic waveforms to focal depth. We first determined moment tensors for the largest earthquakes in the sequence using full waveform inversion, and then modelled high-frequency crustal reverberations in seismograms recorded at a nearby station.

## 2 Data

For this study, we collected digital seismograms for the largest events in the earthquake sequence recorded on all existing stations in the region. This included all broadband stations in Spain including the Balearic Islands, and also short period stations near the CASTOR UGS (Figure 1). The broadband data set consists mainly of stations from permanent networks operated by the Instituto Geográfico Nacional (IGN, network code ES, Instituto Geografico Nacional, 1999) and the Institut Cartogràfic i Geològic de Catalunya (ICGC, network code CA, Institut Cartogràfic i Geològic de Catalunya, 2000). We also benefited from the temporary stations of the TOPO-IBERIA project that were still deployed in northern Spain (Díaz et al., 2009;ICTJA-CSIC, 2007). The short period data set consists of two stations operated by the Ebro Observatory to monitor the seismicity in the vicinity of the UGS (blue triangles in Figure 1).

## 3 Velocity models

Seismic waveforms and earthquake focal depths inferred from them are very sensitive to the Earth's velocity structure. Because the study region is an oil-producing basin, there is a wealth of geophysical information on the structure of the subsurface, including reflection and refraction seismic profiles, seismic velocities, and other petrophysical data obtained from wells. This information is often only available for the upper 2 km where the potential oil bearing formations are located. In addition to this information, mostly vintage in age, a 3-D seismic survey was conducted 2005 in the area of the CASTOR UGS in order to characterize the geometry of the storage and nearby faults (Juanes et al., 2017). Unfortunately,

these data were not available to us, and therefore were not used in this study. In spite of all the existing geophysical data in the region, because the focus was on the shallow structure (i.e. upper 2-3 km), important parameters of the deeper seismic structure such as the total sediment thickness, depth of the crystalline basement, and crustal thickness are relatively poorly known. Constraints on these parameters are provided by the ESCI and other wide-angle profiles (Dañobeitia et al., 1992;Gallart et al., 1994;Vidal et al., 1998), although these were located slightly to the north of the study region (see Figure 1a for location of the ESCI profile).

Because the available information on Earth structure was not adequate for our study, we derived new velocity models for the region. First, we obtained a 1-D model based on surface-wave dispersion measurements and teleseismic P-wave receiver functions at seismic stations near the UGS to represent average wave propagation at distances of 50-650 km. This model was used to compute synthetic low frequency waveforms for moment tensor inversion. Then, another refined 1-D model, also based on surface-wave dispersion combined with well data, was developed to model high frequency waveforms at local distances (less than 40 km). Here we describe in more detail how both models were obtained.

## 3.1 Velocity model for moment tensor determination

Determination of moment tensors in the time domain requires a velocity model that can predict the character of the waveforms in the desired frequency band (0.02 to 0.1 Hz in our case). The requirements on the model are fewer if frequencies lower than 0.02 Hz are used or if observations at short distances are available. However, for small events, signal-to-noise ratio for low frequencies can be low, precluding the use of waveforms to obtain regional moment tensors.

Fortunately, about a dozen of the larger earthquakes in the sequence were well recorded in the Iberian Peninsula. For these earthquakes we measured Rayleigh- and Love-wave group velocities using a multiple filter technique (Herrmann, 1973) that is implemented in the Computer Programs in Seismology (Herrmann, 2013). To these observations we added Rayleigh-wave dispersion estimates (group and phase velocities) obtained from ambient noise tomography. This was done by summing the group and phase delays for each source-station path through the dispersion maps of Palomeras et al. (2017). The purpose of the second step was to obtain additional independent constraints to determine the velocity model, particularly phase velocities of Rayleigh waves (Figure 2b). Combining the dispersion measurements obtained from earthquakes and ambient noise tomography we determined the mean value of group and phase velocity for each period, and used the standard deviation as an estimate of the uncertainty.

Figure 2 shows the obtained dispersion curves with their uncertainties. For Love waves we obtained group velocities from earthquake measurements, and for Rayleigh waves we obtained group velocities from earthquakes and ambient noise, and phase velocities from ambient noise. The standard deviations of the phase velocities are smaller than those of group velocities, and for periods greater than 20 s they are smaller than the symbol size. For Rayleigh wave group velocities there is good agreement between the measurements obtained from ambient noise tomography and from earthquakes. The advantage of the earthquake data is that the dispersion curves can be extended to shorter periods and, in our case, it also

provides Love-wave dispersion measurements (Figure 2a). The derived dispersion curves thus represent an average
propagation velocity to stations in the eastern Iberian Peninsula within about 650 km from the CASTOR UGS.

To create a simple 1-D velocity model to be used for source inversion, we inverted jointly the dispersion data shown in
Figure 2 together with teleseismic P-wave receiver functions for station EMOS (40.36°N, 0.47°W) which is approximately
100 km west of the earthquakes studied (see Figure 1 for location). The joint inversion was performed using the code of
Herrmann (2013). The initial velocity model was the global model ak135 (Kennett et al., 1995), modified in the upper 50 km
to have a constant velocity (that of ak135 at 50 km depth). The purpose of this choice was to have a smooth model that made
no *a priori* assumptions about the sharpness or depth of the Moho. We then simplified the model by combining layers with
similar velocities and truncated it to a depth of 90 km to have a simple velocity model for modeling the waveforms. The
resulting model, denoted VALEN, is given in Table 1.

Figure 3 compares the group velocity dispersion predicted by the VALEN model with predictions from other velocity
models. The group velocities describe the shape of the temporal waveform which is what moment tensor inversion of
waveforms must match. If the velocity model cannot match the observed dispersion, then the inversion suffers (Herrmann et
al., 2011). The other models shown in Figure 3 correspond to two 2° × 2° cells from the global model CRUST2.0 (Bassin et
al., 2000) located in the vicinity of the CASTOR UGS. One of the cells is the one containing the CASTOR UGS (labelled
*offshore* in Figure 3), and the other one is located further inland (labelled *onshore*). Since our moment tensor inversion used
the 16-50 second period range, we can quickly reject the use of the  CRUST 2.0 onshore model. The CRUST2.0 offshore
model could be used, except that the waveform synthetics would still be affected by the very low velocities at short periods.

### 3.2 Velocity model for forward modeling of crustal reverberations

For modeling high frequency body waves, we initially considered the 3-D $v_S$ model of Gaite et al. (2016), evaluated at the
nearest grid point to the CASTOR UGS. In order to reproduce the reverberations recorded at the nearest station ALCN (see
Figure 1 for location), we had to introduce a shallow layer with low velocity that was not resolvable using our surface wave
dataset. Results from marine reflection and refraction experiments in nearby geologic environments similar to our study
region indicate a large velocity contrast between Cenozoic and Mesozoic sediments (e.g. Dañobeitia et al., 1992;Torné et al.,
1992;Vidal et al., 1998). The average depth of the top of the Mesozoic sediments, formed by Cretaceous limestones, is
approximately 2 km in accordance with several borehole stratigraphic columns in the area. Therefore, we added to the top of
our model a 2-km thick layer with a *P*-wave velocity of 2.4 km/s, representative of the Cenozoic sediments. The velocity
value of this first layer is selected from results of refraction and wide-angle reflection profiles recorded with OBS and land
stations that cross the continental platform north of the Ebro Delta (Profile I in Dañobeitia et al., 1992). This velocity is
lower than the average value obtained from velocity logs closer to the area ($v_P$ around 2.8-3.0 km/s for the first 2 km from
Castellon C-3 well), however it fits better the observed waveforms. The complete 1-D model ($v_P$, $v_S$, density, *P* and *S*
attenuation) used to compute high-frequency ground motion was constructed considering a $v_P$ /$v_S$ ratio of 1.75, the density-

velocity relationship $\rho = 0.32\ v_P + 0.77$ (Berteussen, 1977), a $Q_S$ value of 100 (Ugalde et al., 1999), and $Q_P = 0.76\ Q_S$ (Mancilla et al., 2012) (Table 2).

## 4 Focal mechanisms from waveform inversion

We analyzed all earthquakes in the IGN catalog with reported magnitudes $m_{bLg} \geq 3.5$. From all the events studied we obtained reliable mechanisms for 14 earthquakes with $M_w$ ranging between 3.0 and 4.1 (Table 3).

The waveform inversion method used here is described in detail by Herrmann et al. (2011) and will only be briefly summarized. Three-component waveforms were converted to velocity and rotated to radial, transverse and vertical components. Next the seismograms were bandpass filtered between 0.02 and 0.06 Hz (16 – 50 s) to evaluate their quality. We selected waveforms that showed retrograde motion for the fundamental model Rayleigh wave, good signal to noise ratio, and finite signal duration.

The inversion method uses a grid search approach that samples over strike, dip and rake angles in 5° increments, and source depth in 1 km increments, in order to determine the shear-dislocation (double couple) that best fits the observed data. A feature of the implementation of the grid search is an efficient method for adjusting the predicted waveforms for time shifts that arise because of uncertainties in the assumed origin time and epicentral coordinates, the sampling of the Green's functions with distance, and differences between the actual wave propagation and that of the 1-D model used.

Since the largest signals observed in the frequency band used for inversion are surface waves, and since the initial P-wave signal usually fades into background noise at larger distances, we used a window that extended from 30 seconds before to 60 seconds after a group velocity arrival of 3.3 km/s. Finally, we filtered both the observed and Green's function ground velocities by applying a 3-pole highpass Butterworth filter at 0.03 Hz an a 3-pole lowpass filter at 0.06 Hz. For the larger events, we used a highpass filter at 0.02 Hz, and for small events a lowpass at 0.1 Hz. The objective of the filtering was to use as wide a frequency range as possible, to have a good signal-to-noise ratio, and yet to use low frequencies so that errors in the 1-D velocity model would be minimized. Although there are mixed water-land paths to the stations, the 1-D model is assumed adequate because water depth is small (maximum of ~60 m), and most of the paths are continental. We searched source depths from 1 to 29 km in increments of 1 km to represent depth below the base of the water.

As an example of the processing, we present the results for the largest event, the $M_w$=4.08 earthquake of 2013-10-01 at 03:32 UTC. Figure 4a shows the location of the event and the stations used for the source modeling. The data set has an epicentral distance range from 50 to 650 km and covers an azimuth range slightly over 180 degrees. Unfortunately, many of the stations share similar azimuths and thus provide redundant information. Figure 4b presents the observed and predicted waveforms for the optimal solution at selected stations at distances between 70 and 405 km. The low frequency part of the signals is modeled fairly well, as are some of the earlier P-wave arrivals. We do not expect the fits to be perfect given the variability of structure from the source region to the individual stations. The fits are judged adequate on the basis of the relatively small time shifts and because of the low frequency used. The waveform comparison shown in Figure 4b indicates

an excellent fit to the transverse component at EORO while the corresponding vertical and radial components are not as well fit because this station is near a minimum (nodal plane) of the radiation pattern. The difference in the durations of the Rayleigh wave and the Love wave at CBEU reflects the difference in the dispersion curves – the Rayleigh wave group velocities flatten out a bit in Figure 2, which gives rise to a pulse in the synthetics and observed seismograms.

Figure 5 presents the best fitting solution as a function of source depth for two different frequency bands. Our best solution for the frequency band 0.03-0.06 Hz (Figure 5a), which is suitable for most of the events analyzed, has a source depth of 7 km, a moment magnitude of 4.08, and strike, dip and rake angles of 40, 55 and -5, respectively. The data fit is relatively good, but does not show a sharp peak in depth, but rather a broad maximum between 4 and 12 km. Although the uncertainty in depth is high, we can certainly reject depths less than 3 km or greater than 15 km. Although not indicated on the plot, the estimated moment magnitude increases with depth because the material properties increase with depth in the model.

When we extended the frequency band from 0.03-0.06 Hz to 0.02-0.1 Hz, the goodness of fit was slightly reduced, but the source depth peaked more sharply at the slightly deeper depth of 9 km (Figure 5b). We repeated this exercise for the three largest events, and found that in all cases the higher frequency band led to a source depth of about 2 km deeper with a sharper indication of depth.

Table 3 summarizes the source parameters determined in this study (epicenters are taken from Gaite et al. (2016)). In Figure 6a we show the focal mechanisms for all the earthquakes that we were able to process successfully. Most of them correspond to strike-slip mechanisms with a small component of normal faulting. Almost all the events exhibit a well-constrained near-vertical nodal plane that strikes NW-SE, with more variability in the orientation of the other nodal plane. Figure 6b shows the orientation of the P axis of the focal mechanisms, which is predominantly N-S. The only exception is the easternmost event (#1 in Table 3), which occurred in 2012-04-08 before the beginning of the injection activities at the CASTOR UGS. In Figure 6b we have plotted the orientation of the P axis, color-coded according to the relative proportions of thrust, strike-slip and normal component of the mechanism (Frohlich, 1992). Most of the mechanism have a proportion of 60% or more of strike slip motion (shown as green bars in Figure 6b), while the rest do not have a predominant component (grey bars). In Figure 6c we show measurements and the average direction of the maximum compressive stress axis $S_{Hmax}$ (see Zoback, 1992) in the region of the CASTOR UGS according to the recent update of the World Stress Map (Heidbach et al., 2016). The calculated average direction stress of $S_{Hmax}$ (grey bars), and the measurements from borehole data (Schindler et al., 1998) coincide extremely well with the orientation of the P axis of the focal mechanisms obtained for the largest earthquakes in the sequence.

## 5 Modeling of short period crustal reverberations

A noticeable feature in the short-period seismograms recorded at short distances are several relatively high amplitude phases arriving after the direct *S* phase, clearly observed on the transverse component (Figure 7). We interpret these arrivals as crustal reverberations generated when the source is near a velocity boundary, and significant amounts of energy are trapped

in the uppermost layers. The amplitude and temporal separation of these reverberation phases is very sensitive to focal depth so, by modeling them we expect to obtain additional constraints on the focal depths of the largest earthquakes in the sequence. We modeled these ground motion displacements using the program FK (Zhu and Rivera, 2002), following the approach described in Frohlich et al. (2014).

We computed the synthetic ground motion generated by the largest earthquakes of the sequence at the closest station location
(ALCN), at approximately 15 km distance (Figure 1). We used the epicentral locations calculated by Gaite et al. (2016) obtained using a 3-D model, and the seismic moment tensor solutions computed in the previous section from full waveform inversion. As velocity model, we considered the 1-D model based on ambient noise tomography combined with well data described in section 3.2 and listed in Table 2.

We computed synthetic seismograms of the transverse component of ground displacement for focal depths varying from 1 to
22 km in 1 km increments. Both the synthetic and observed seismograms were band-pass filtered between 0.2 and 2 Hz and integrated to displacement for comparison. To measure the goodness of the fit we calculated the cross-correlation coefficient between the observed and synthetic seismograms. The most likely focal depth was chosen as the one that provided the largest value of the cross-correlation coefficient between the observed and synthetic seismograms.

For all the earthquakes analyzed the focal depths that resulted in a higher cross-correlation coefficient were in the range
between 6 and 8 km. (Figure 7). This is in accordance with the average ~ 6 km depth obtained by Gaite et al. (2016) using a 3-D velocity model and refined picks using waveform similarity.

**6 Discussion**

We will now discuss the implications of our results (focal mechanisms and focal depths) for the identification of the faults reactivated during this episode of induced seismicity and the process responsible for this reactivation.
We will first examine the similarities and differences of our results with previous studies. Cesca et al. (2014) performed the first seismological study of this earthquake sequence. They used catalogued arrival times, a global regionalized velocity model (CRUST2.0), and long-period spectral amplitudes to solve for the moment tensor and focal depth. Their results differ from ours in several ways. For the 12 events in common in both studies, their depths are shallow (1 to 2 km depth), their moment magnitudes are about 0.2 $M_w$ units greater, and although one nodal plane is in the NW-SE direction, the other nodal
plane dips very shallowly to the southeast. Differences might be caused by the model used (CRUST2.0 vs. our local model) and the type of data (spectral amplitudes vs. full waveforms).

Saló et al. (2017), using the waveform inversion approach of Delouis (2014), obtain mechanisms similar to ours, predominantly strike-slip, with one near-vertical nodal plane striking NW-SE, and a second nodal plane dipping to the SW. Their focal depths are also similar to ours (mostly 5-8 km depth).
Recently Juanes et al. (2017) have also obtained locations and focal mechanisms for the events in this sequence. Using a 1-D Earth model and catalogued arrival times, they obtain focal depths generally shallower than 5 km. However, when using a 3-

D velocity model derived from their 3-D structural model, they obtain deeper focal depths, between 5 and 15 km, in agreement with the results of Gaite et al. (2016). This is not surprising since their detailed 3-D structural model in the vicinity of the CASTOR UGS was embedded in the regional model of Gaite et al. (2016). Their focal mechanisms, obtained

using waveform fitting (Li et al., 2011), are also predominantly strike-slip with a steeply-dipping NW-SE nodal plane, and a vertical SW-NE nodal plane. In their report however, they do not provide estimates of focal depth obtained from waveform fitting.

The discrepancies between these studies are, in our view, more representative of the poor configuration of the monitoring network of the CASTOR UGS, than of the complexity of the structure in the region or the variability of the earthquake

sources. Data from one or more ocean-bottom seismometers in the vicinity of the storage would have allowed to discriminate between shallow (1-2 km) and deeper (> 5 km) focal depths with very small uncertainty. Lacking data from reliable, nearby stations, errors in epicenter and focal depth can be too large to allow for a confident association of the seismicity to a specific fault or faults. Gaite et al. (2016) attempted to decrease the location uncertainty by creating a 3-D velocity model of the region, and by obtaining precise arrival time picks exploiting the similarity of waveforms from nearby earthquakes. Using

this approach, they obtained a distribution of epicenters with a predominantly NW-SE orientation, and focal depths generally > 5 km. Juanes et al. (2017) also obtain a NW-SE orientation of the epicenters, and deeper (> 5 km) focal depths when using their 3-D model, while using a 1-D model results in shallower (< 5 km) focal depths. On the other hand, Cesca et al. (2014), using the 1-D model in CRUST2.0 (Bassin et al., 2000) for the source region, and a waveform coherence location method (Grigoli et al., 2014) obtain very shallow locations, and an approximately N-S distribution of epicenters (see their Figure 6).

Interestingly, the best constrained and therefore more consistent feature of all the focal mechanisms obtained for this sequence is the near-vertical NW-SE striking nodal plane. This coincides with the epicenter distribution obtained by the IGN, Gaite et al. (2016), and Juanes et al. (2017). However, there is no major know active fault in the region with this orientation. The predominant orientation of active faults in the Gulf of Valencia coast is SW-NE (Garcia-Mayordomo et al., 2012) with the exception of some minor faults that splay off from main Amposta fault to the east (grey lines in Figure 5a,b).

These faults shown in Figure 5 were obtained by Geostock (2010) from the analysis of recent, more detailed 3D seismic studies carried out to delineate the reservoir size.

In addition to the distribution of epicenters, another important parameter to help identify causative faults is focal depth. Fortunately, the poor constraints provided by arrival time data to focal depth in absence of nearby stations are compensated by the large sensitivity of seismic waveforms to depth. By performing waveform inversion to obtain source parameters

(depth, scalar moment, and focal mechanism), and by modeling high-frequency reverberations of $S$ waves, we obtained strong constraints on focal depth. Using both approaches we determined optimum depths centered at around 6-8 km depth. The uncertainty of these estimates, provided by the shape of the fitting curve (e.g. Figure 5 and right panels in Figure 7) is relatively large, but for both approaches depths shallower than 4 km provide a poor fit to the waveform data. Saló et al. (2017) using a waveform inversion approach also obtain deeper focal depths (5-8 km), while Cesca et al. (2014) fitting

amplitude spectra obtain shallow focal depths (2 km). When a good distribution of recording stations is available, waveform

inversion methods should provide better sensitivity to focal depths than those based on spectral amplitudes. Also, using a velocity model that more accurately predicts the characteristics of waveform propagation in the region should provide more reliable results. This, combined with the good fit of short period reverberations obtained in the previous section leads us to propose that the larger events occurred at depths of 5-8 km, significantly greater than the injection depth of ~2 km. This scenario is very frequent for fluid-injection induced earthquakes, where the seismicity occurs in the crystalline basement, and not in the sedimentary layers where the injection takes place (e.g. McNamara et al., 2015).

The association of the obtained nodal planes to causative faults of the earthquakes presents also some difficulties for this sequence. Cesca et al. (2014) do not favor any of their two nodal planes (shallow dipping to the SE, and steeply dipping striking NW-SE), and propose two potential scenarios of fault reactivation. Their analysis also excludes the reactivation of the Amposta fault. On the other hand, Juanes et al. (2017) propose the reactivation of the Amposta fault, although none of the nodal planes in their mechanism dips to the NW. In all the studies reviewed here, there is not a single focal mechanism that presents a W- or NW-dipping nodal plane corresponding to the geometry of the Amposta fault in the region (which dips 40-60° to the NNW according to Figure 2.2 in Juanes et al. (2017)).

Although the deep structure in the region of the CASTOR UGS is not known in great detail, a depth of 6 km is most likely deeper than the Cenozoic and Mesozoic sediments, and within the Hercynian (Paleozoic) extended crust beneath the Iberian margin. We will refer to this layer as the crystalline basement. The extended crust beneath this segment of the Valencia trough was accommodated by a listric normal fault system reaching detachments depths of up to 15 km depth (Roca and Guimerà, 1992). This fracture network could have acted as a high-permeability pathway for pore-pressure perturbations to reach the crystalline basement and trigger faults that were critically stressed. An alternative mechanism for induced earthquake triggering could be aseismic fault slip. Using fluid-injection experiments on shallow crustal faults, Bhattacharya and Viesca (2019) show that aseismic fault slip can transmit stress changes faster and to larger distances than pore-fluid migration. Considering that small-magnitude induced earthquakes began to occur 2 days after the injection of the base gas started, and that the largest earthquakes occurred only 4 weeks later (and 2 weeks after the end of the injection), aseismic fault slip (for example at the Amposta fault) could be a viable mechanism for triggering the sequence. However, without detailed studies of geomechanical modeling, this assertion remains speculative.

Although the nodal planes of the focal mechanisms obtained for the CASTOR sequence are not consistent with the orientation of any of the main faults and structures imaged in the region of the storage, faults in the crystalline basement might have different orientation than those in shallow layers. It is not uncommon that old unmapped faults in the basement that have not shown previous seismic activity are reactivated by the injection of fluids (e.g. Yeck et al., 2016;Keranen and Weingarten, 2018). During the Middle Jurassic, the region immediately west of the CASTOR UGS was transected by a complex network of NW- and NE-trending faults (Gómez and Fernández-López, 2006), some of which could have been reactivated by the gas injection. In particular the seaward continuation of the NW-trending Vinaros fault would be compatible with the NW-SE nodal planes of the focal mechanisms obtained.

In view of the evidence presented here, we postulate that the large earthquakes in this sequence occurred in faults in the crystalline basement. We favor the NW-SE striking nodal plane as fault plane because it coincides with the distribution of seismicity. However, we cannot discard the SE dipping nodal plane, because it coincides with the orientation of mapped faults that affect the Cenozoic and Mesozoic sediments, and presumably also could affect the crystalline basement.

Based on our consistent results of focal depths in the range of 6-8 km using different approaches, and in the absence of nodal planes compatible with the Amposta fault, we consider that it is unlikely that the largest earthquakes in this sequence occurred on the Amposta fault.

## 7 Conclusions

In this study, we have obtained new source parameters (focal depths and mechanisms) for the largest earthquakes in the CASTOR sequence using full waveform inversion. The focal depths obtained range between 5-10 km, consistent with results from the modeling of crustal reverberations, that provide a narrower depth range (6-8 km). These depths indicate that the reactivated faults are located in the crystalline basement, significantly deeper than the injection depth (~ 2 km).

Focal mechanisms correspond to strike-slip motion with a small normal fault component. The orientation of the maximum compressive stress $S_{Hmax}$ derived from these earthquakes is N-S, in good agreement with the regional stress regime, indicating that these earthquakes occurred in critically stressed faults subject to regional stresses. None of the nodal planes obtained by this or other studies is compatible with reactivation on the Amposta fault.

In spite of our analysis, uncertainties still remain with respect to the focal depth of the earthquakes and the causative fault. This is mainly due to the poor configuration of the seismic network deployed to monitor this facility, particularly the lack of seismometers on the ocean bottom (OBSs) and in the observation wells.

*Data availability*. The seismic data used in this study, and the obtained regional centroid moment tensor solutions (including information on the velocity model and stations used for each solution, focal depth sensitivity, data fit, et cetera) is publicly available in this data repository: https://digital.csic.es/handle/10261/192082 (Villaseñor et al., 2019).

*Competing interests*. The authors declare that they have no conflict of interest.

*Acknowledgements*. We thank the seismic networks that provided the waveform data used in this study: IGN (https://doi.org/10.7914/SN/ES), and ICGC (https://doi.org/10.7914/SN/CA). This research was funded by project SEAL (Ministerio de Ciencia e Innovación, Spain, CGL2017-88864-R).

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

**Table 1.** Velocity model for moment tensor determination

| Layer thickness (km) | $v_P$ (km/s) | $v_S$ (km/s) | density (g/cm$^3$) | $Q_P$ | $Q_S$ |
|---|---|---|---|---|---|
| 2 | 3.54 | 1.97 | 2.24 | 330 | 150 |
| 2 | 5.38 | 3.00 | 2.57 | 330 | 150 |
| 8 | 6.11 | 3.41 | 2.73 | 330 | 150 |
| 2 | 6.28 | 3.50 | 2.78 | 450 | 200 |
| 12 | 6.53 | 3.64 | 2.86 | 450 | 200 |
| 12 | 7.35 | 4.10 | 3.09 | 450 | 200 |
| 8 | 7.83 | 4.37 | 3.25 | 900 | 400 |
| 5 | 7.74 | 4.32 | 3.22 | 900 | 400 |
| 20 | 7.80 | 4.35 | 3.24 | 900 | 400 |
| 15 | 7.97 | 4.45 | 3.30 | 900 | 400 |
| halfspace | 8.07 | 4.50 | 3.33 | 2250 | 1000 |

**Table 2.** Velocity model for forward modeling of crustal reverberations

| Layer thickness (km) | $v_P$ (km/s) | $v_S$ (km/s) | density (g/cm$^3$) | $Q_P$ | $Q_S$ |
|---|---|---|---|---|---|
| 2 | 2.40 | 1.37 | 1.54 | 100 | 76 |
| 3 | 4.79 | 2.74 | 2.30 | 100 | 76 |
| 11 | 5.78 | 3.30 | 2.62 | 100 | 76 |
| 38 | 7.35 | 4.20 | 3.12 | 100 | 76 |
| 90 | 7.80 | 4.46 | 3.27 | 100 | 76 |

**Table 3.** Source parameters obtained in this study for the largest earthquakes in the vicinity of the CASTOR gas storage

| # | Date | Time | Latitude (°) | Longitude (°) | Depth (km) | Mw | strike (°) | dip (°) | rake (°) |
|---|---|---|---|---|---|---|---|---|---|
| 1* | 2012-04-08 | 11:58:44 | 40.339 | 0.775 | 6.0 | 3.20 | 20 | 90 | -40 |
| 2 | 2013-09-24 | 00:21:50 | 40.401 | 0.677 | 9.0 | 3.50 | 45 | 55 | -5 |
| 3 | 2013-09-25 | 05:59:49 | 40.382 | 0.711 | 9.0 | 3.05 | 230 | 50 | 30 |
| 4 | 2013-09-29 | 16:36:23 | 40.374 | 0.722 | 8.0 | 3.46 | 230 | 55 | 10 |
| 5 | 2013-09-29 | 21:15:06 | 40.389 | 0.720 | 10.0 | 3.25 | 45 | 55 | 10 |
| 6 | 2013-09-29 | 21:23:16 | 40.374 | 0.689 | 5.0 | 3.11 | 40 | 60 | -30 |
| 7 | 2013-09-29 | 22:15:48 | 40.378 | 0.715 | 7.0 | 3.63 | 40 | 55 | -5 |
| 8 | 2013-09-30 | 02:21:16 | 40.375 | 0.706 | 8.0 | 3.84 | 45 | 60 | 0 |
| 9 | 2013-10-01 | 03:32:44 | 40.378 | 0.742 | 7.0 | 4.08 | 40 | 55 | -5 |
| 10 | 2013-10-02 | 23:06:49 | 40.380 | 0.718 | 4.0 | 4.01 | 40 | 70 | -5 |
| 11 | 2013-10-02 | 23:29:29 | 40.413 | 0.678 | 7.0 | 3.97 | 35 | 60 | -5 |
| 12 | 2013-10-04 | 08:49:48 | 40.408 | 0.659 | 9.0 | 3.69 | 40 | 70 | -15 |
| 13 | 2013-10-04 | 09:55:19 | 40.373 | 0.724 | 4.0 | 3.43 | 35 | 75 | 0 |
| 14 | 2013-10-04 | 20:02:24 | 40.369 | 0.727 | 10.0 | 3.47 | 30 | 35 | 0 |

* This event occurred before the 2013 seismic sequence


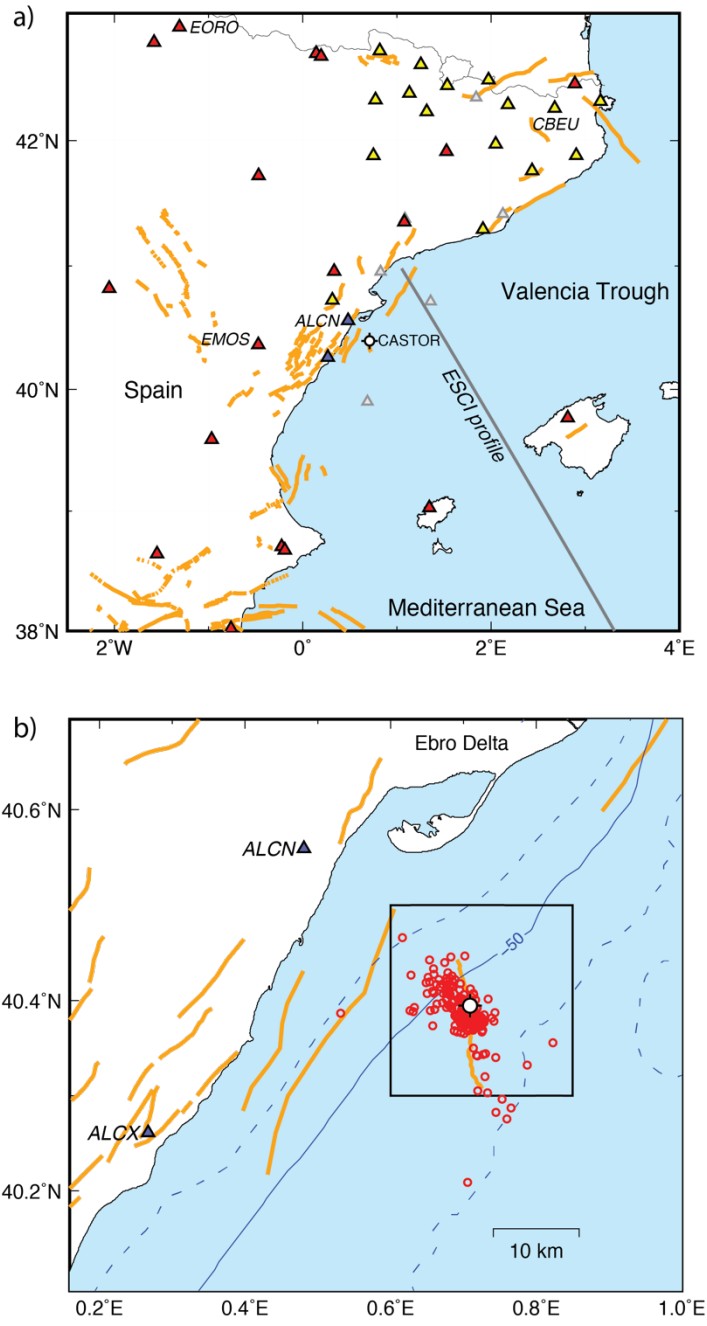

**Figure 1. a) Location of the CASTOR UGS (white circle) and permanent seismic stations in the study region. Blue triangles: Ebro Observatory (network code EB); red triangles: IGN (ES); yellow triangles: ICGC (CA); grey triangles: permanent stations not used (not available, not operating at the time, or with instrumentation problems). Station codes of stations cited in the text are labeled. b) Zoom in the region of the CASTOR UGS showing bathymetry in meters (dashed lines are every 25 m), and earthquake locations of the 2013 sequence (red circles), relocated by Gaite et al. (2016). The black box indicates the region shown in Figure 5.**

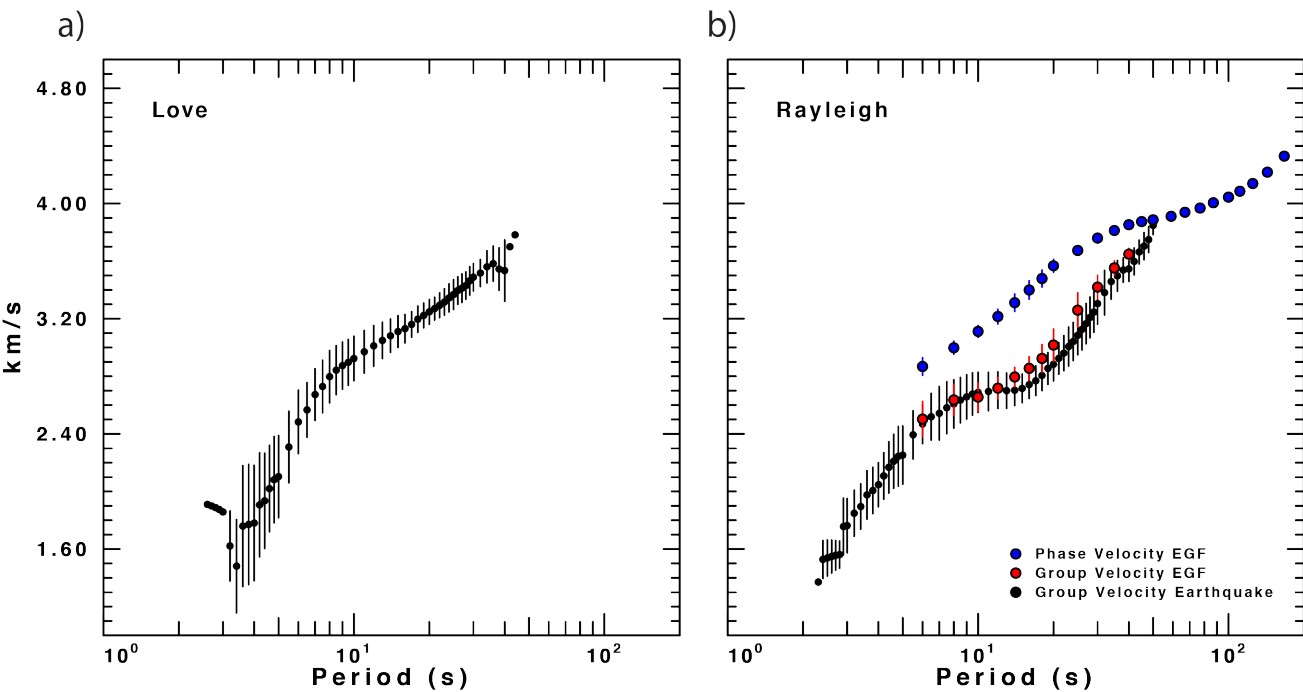


**Figure 2. Dispersion measurements used to obtain the VALEN 1-D model for waveform inversion. Group velocities from earthquakes are shown as black circles, group velocities from noise as red circles, and phase velocities from noise shown as blue circles. Vertical error bars indicate measurement uncertainty (standard deviation). For phase velocities some error bars are smaller than the symbol size. a) Love wave dispersion measurements, and b) and Rayleigh wave dispersion measurements.**


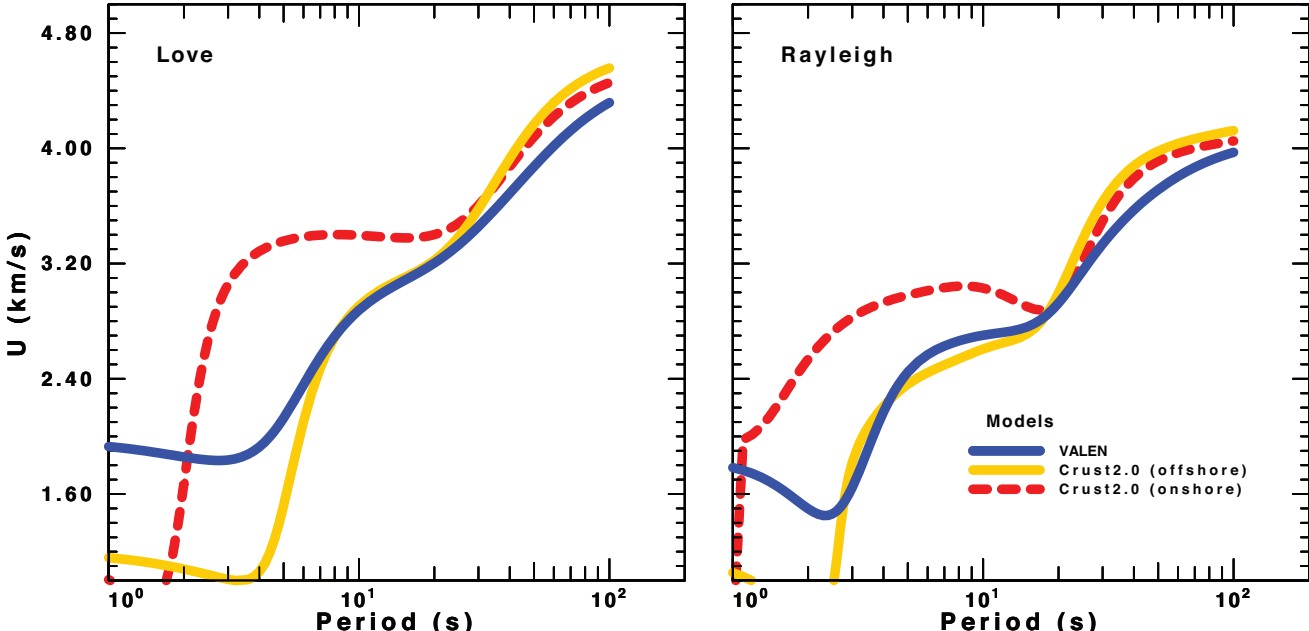

**Figure 3. Predicted group velocities for different models discussed in the text. VALEN is the model used for waveform inversion, CRUST2.0 offshore is the grid point of CRUST2.0 closest to the CASTOR UGS, and CRUST2.0 onshore is the grid point immediately to the west, located in the eastern Iberian Peninsula. Love wave group velocities are show on the left panel, and Rayleigh wave group velocities on the right.**

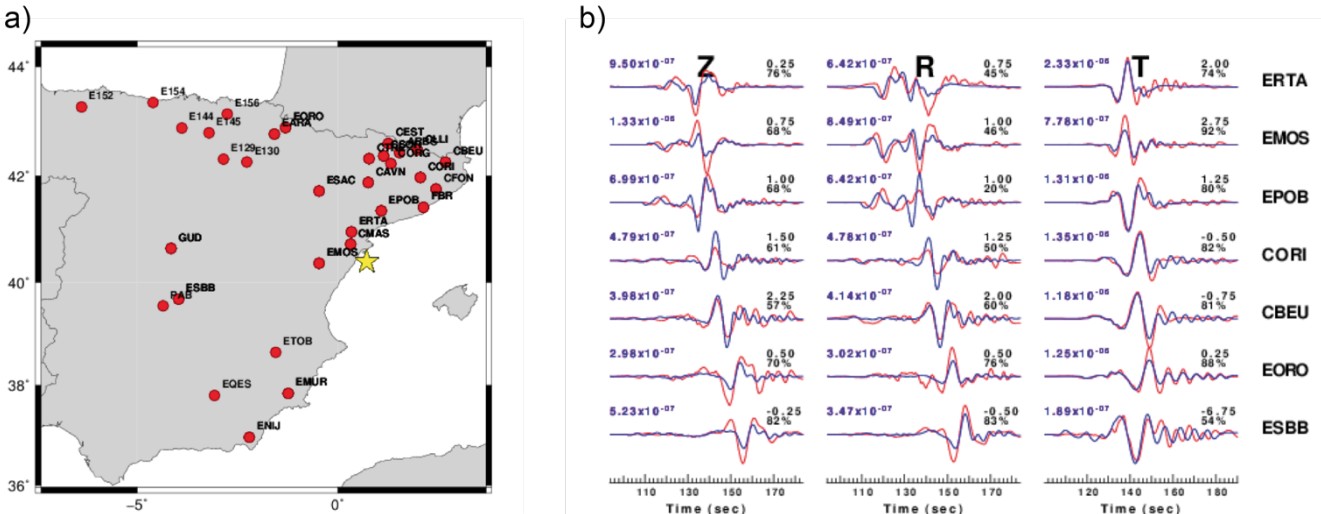

**Figure 4. Regional moment tensor determined for the largest earthquake of the sequence, occurred on 2013-10-01 03:32 UTC, with $M_w = 4.08$. a) Location of the earthquake (yellow star) and of broadband stations that were used to determine this moment tensor (red circles). b) Waveform fits for the optimum solution of the moment tensor for this earthquake. Z indicates vertical component, R radial, and T transverse. Observed (red) and predicted (blue) ground velocities for the optimum solution are shown for 7 selected stations.**

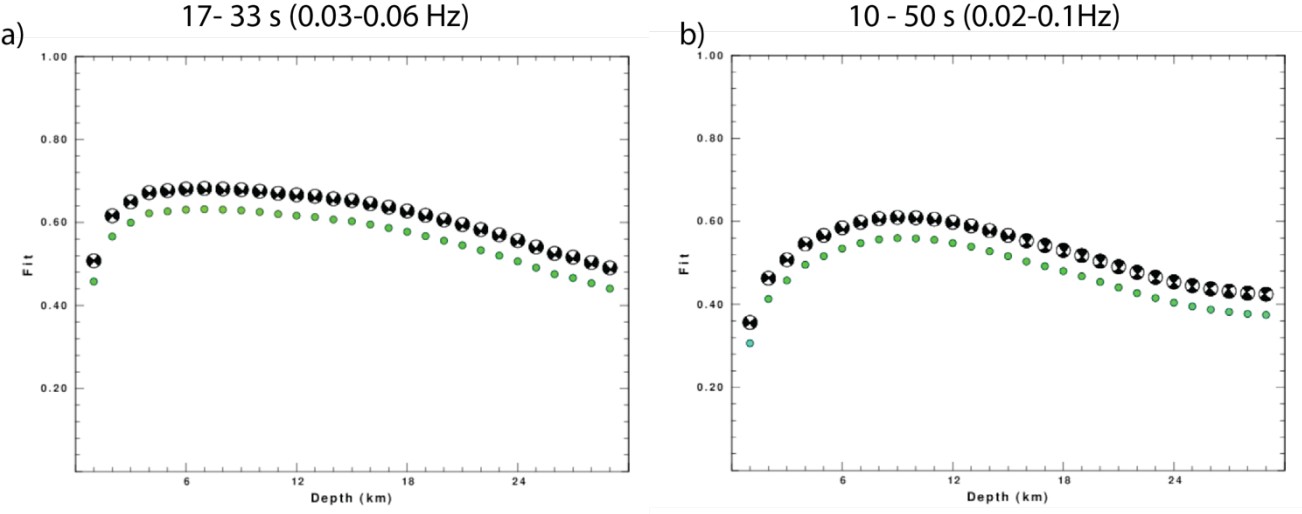


**Figure 5. a) Normalized goodness of fit versus focal depth for the earthquake shown in Figure 4 (2013-10-01 03:32 UTC, $M_w$ = 4.08.) using a frequency band of 0.03-0.06 Hz. Perfect fit corresponds to a value of 1. For each depth, the best-fitting focal mechanism is shown. b) Same as a) but for the frequency band of 0.02 to 0.1 Hz.**


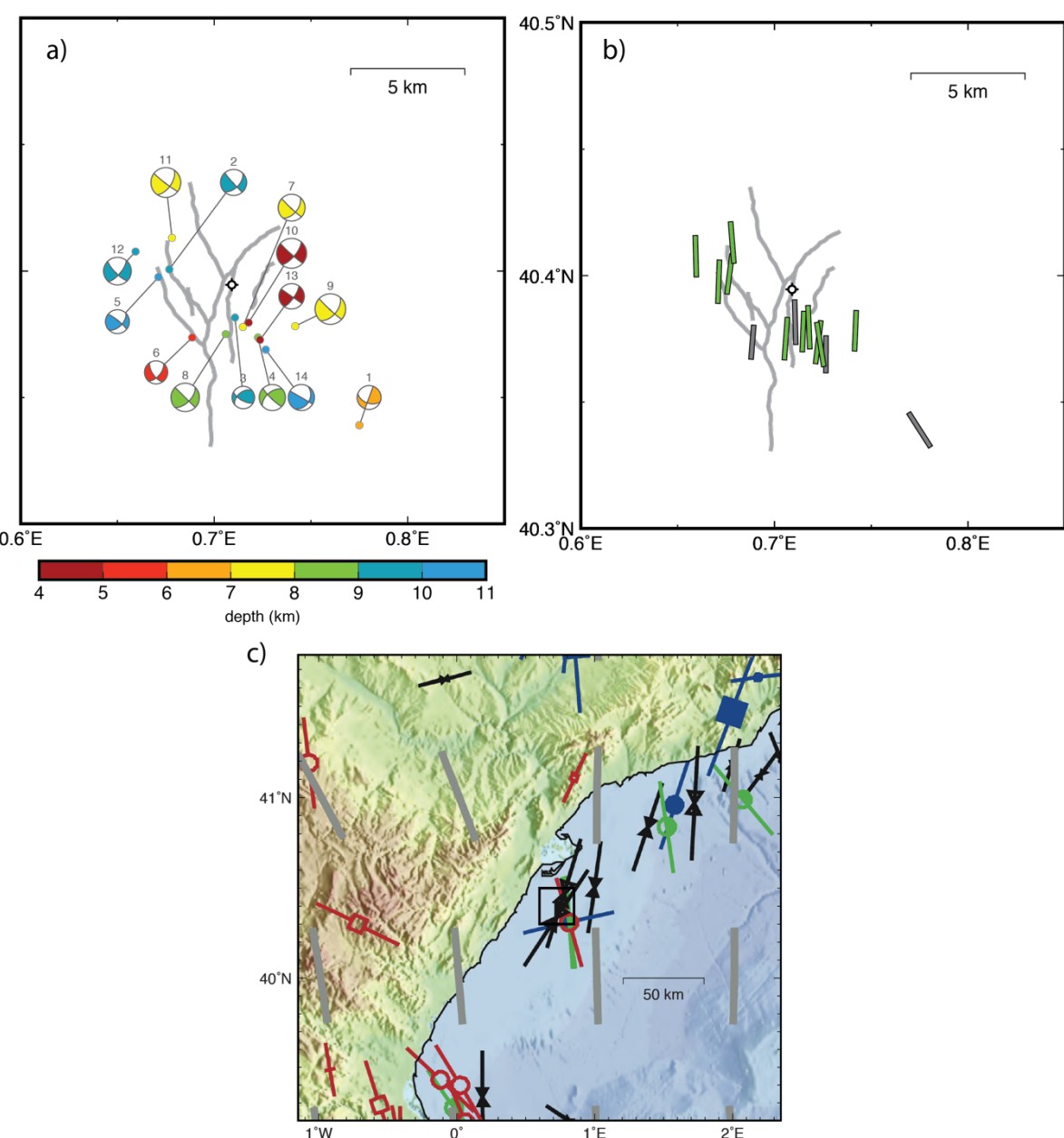

**Figure 6.** Focal mechanisms obtained in this study. a) Nodal planes projected in the lower hemisphere of the focal sphere. Colored quadrants correspond to compression, and the color represents focal depth according to the legend. Numbers above each beach ball correspond to the solution listed in Table 3. Thick grey lines indicate the traces of main faults in the area at 1700 m depth (Geostock, 2010) b) Orientation of the P axes of the mechanisms shown in panel a). Green indicates predominantly strike-slip mechanism, and grey mixed type. c) Stress measurements and mean $S_{Hmax}$ orientations in the region of the CASTOR UGS from the current update of World Stress Map (Heidback et al., 2016). Grey bars are the mean $S_{Hmax}$ orientations on a 1° grid estimated with a 250-km search radius and weighted by data quality and distance to the grid point. For other symbols see the legend in Heidback et al. (2016).

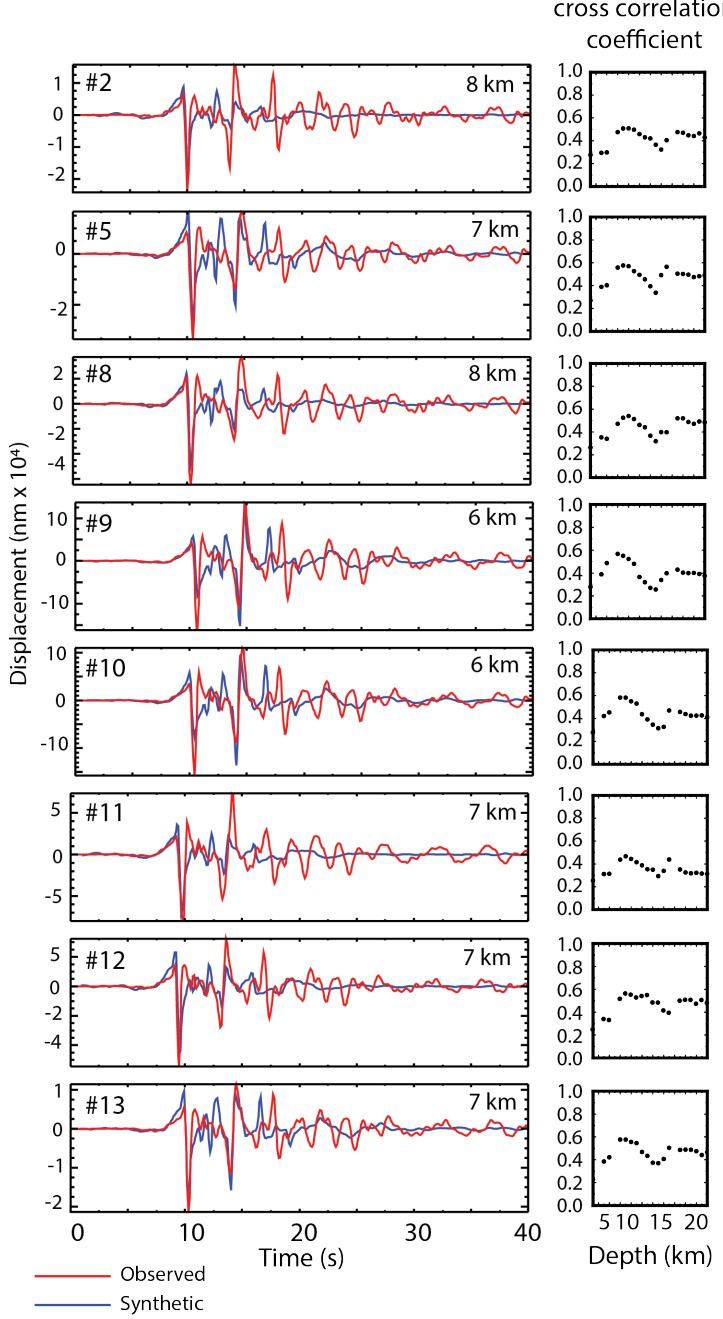

**Figure 7. Transverse component of the *S*-wave ground displacement (in nm) for 8 of the largest earthquakes in the sequence recorded at station ALCN (see Figure 1 for location). Red lines are the observed data, and blue lines are the synthetic waveforms for the best fitting depth. Event number according to Table 3 is indicated in the upper left of each seismogram, and best fitting focal depth in the upper right. The right panels show the cross-correlation coefficient between the observed and synthetic displacement seismograms as a function of depth. All events show low cross-correlation values for shallow depths (less than 2-4 km) and a clear maximum between 6-8 km.**


