# Peer review of "Fault reactivation by gas injection at an underground gas storage off the east coast of Spain"

_Solid Earth, 2019_

## Referee Comment (RC1) · Heather DeShon (Referee) · 29 Jul 2019

General Comment: The manuscript "Fault reactivation by gas injection at an underground gas storage off the east coast of Spain" by A. Villasenor, R. Hermann, B. Gaite and A. Ugalde presents improved moment tensor solution for moderate magnitude earthquakes associated with induced earthquakes occurring offshore Spain in 2013. The motivation was to resolve a depth discrepancy for the earthquakes, which currently exists in the literature regarding the event sequence, in order to better understand the causal link between the gas storage facility, faults, and triggered seismicity. The study provides a careful analysis of moment tensors constrained using surface wave data and crustal reverberations to conclude that earthquake depths were between 6-9 km below the surface, in line with reactivation of presumed pre-existing NW-SE trending

basement faults, rather than the 2 km depth in other papers consistent with injection levels. The study hypothesizes that to pore fluid pressure diffusion away from injection changes stress on the pre-existing fault structure enough to induce primarily strike-slip earthquakes consistent with the modern stress regime, in line with current research on induced earthquakes in Oklahoma, USA, for example. The paper requires minor changes to the text and figures to ensure consistency.

Specific Comments/Questions: The authors favor pore fluid pressure diffusion to link injection at <2 km to faulting at >6 km. The authors establish that there is a lack of geologic information for the crystalline basement in the study area publicly available. Is there any indication in the literature that faults in the basement offset the overlying units or that there is an extensive fault or fracture network that could serve to rapidly transmit fluid pressure? Lines 384-385 hypothesize that faults in the basement have a different orientation than faults in the shallow geologic formations. Is there any evidence from the regional data that this could be the case? Are there any faults that could be added to the figures to aid the reader in understanding the overall geologic setting? On lines 341-344 the authors reference faults as plotted in other studies but could these be added to the figures here for clarity? Is triggering via poroelastic stress change necessary?

Supplementary Material: It was not clear to me why the information in the supplement (1 paragraph essentially and 1 figure) was not included in the main text. It seemed a valid question worth addressing in the main text. I leave it to the authors' decision however.

Citations: In addition to Yeck et al. and McNamara et al., this paper could cite a review paper such as Keranen and Wiengarten (2018), Induced Seismicity, Annual Review of Earth and Planetary Sciences, Vol. 46:149-174, https://doi.org/10.1146/annurev-earth-082517-010054

Figures: In general, the graphics clearly illustrate the points made in the main text. The

fonts on the legends are very small, however. There is also a change in color scheme for data vs modeled waveforms in the main text and supplemental figure; red should be consistently used for modeled waveforms with blue/black used for observed data.

In Figure 2, the size of the circles make it difficult to tell the difference between EGF phase and group velocity (though of course the offset in c/U makes this clear).

In Figure 5, the color bar is marked incorrectly. For example, red is 4 but having the 4 on the far left such that both 4 and 5 bound the red in the color box is not correct. This ends up making 9 and 10 km depth the same color, though there are at least 2 earthquakes at 10 km depth. Most importantly, what are the grey anastomosing lines? They are not referenced in the text or the caption for the figure.

In Figure 6, the open circles and font sizes associated with the cross-correlation column are too small. The open circles can just be made solid, which may solve the small line width issue.

---

## Referee Comment (RC2) · Anonymous Referee #2 · 23 Aug 2019

Review, Solid Earth, se-2019-113 Villasenor, A., et al. "Fault reactivation by gas injection at an underground gas storage off the east coast of Spain"

The paper provides a seismological discussion on an interesting case of triggered seismicity in Europe, occurring in 2013 offshore Spain. The sequence was studied by a number of previous publications and reports. However, beside a general agreement on the relatively shallow hypocenters and strike-slip dominated mechanisms, accurate depth and fault geometry remain to a certain extent debated. Given the interest of the sequence and its relevant in the field of induced seismicity, this study appears to be justified.

Target of the study are basically on one side dispersion curves and velocity models, to improve Green's function and data modeling up to higher frequencies, and on the other

side a contribution to the estimate of focal depths and focal mechanisms (or moment tensors).

I think this is an interesting manuscript, but requires some moderate improvement. I provide below my major comments:

Main comments:

1. Uncertainties

In order to provide new insights into a sequence which was discussed by previous papers, I think authors should not only provide a new result (depth, location or mechanism) but also some uncertainties. The estimation of uncertainties is discussed indeed in the first sections, dedicated to the assessment of dispersion curves and velocity models, but they are not used to derive a uncertainties on derived parameters, such as the depth.

2. Network asymmetry

Both depth estimation, location and hypocenters suffer in this region by the asymmetric distribution of the stations. In this study, some new data have been taken into account (e.g. upon the Topoiberia project), but the azimuthal coverage remain strongly unbalanced. This may have a strong influence on the location accuracy, and some works suggested that the distribution plotted e.g. in Fig. 1b, may be partially attributed to the network geometry. The azimuthal coverage may also affect the depth, because of an inaccurate epicentral location. Has this been verified? Finally, it surely affects the focal mechanisms estimation. All these effects are not discusses.

3. Data used for MT inversion

Furthermore, authors use the same velocity model for all stations. While this may be proper for onshore stations, I doubt this is accurate for stations on Balearic islands. It is unclear whether these stations have been used or not, as they appear in Fig. 1 but not in Fig. 4. Using them will surely improve the coverage, and improve the moment

tensor inversion result, but possibly a different velocity model should be used. Fig.4c should show some waveform fit there.

4. Velocity models

Since a lot of velocity models are discussed, they should be included in the document, as table or in the e-supplement. Having them available is need for the reproducibility of results

5. High frequency waveform modeling

The high-frequency waveform comparison is very interesting and in my opinion the most interesting and novel part of the work. However, too little is said on how data were processed. Please, provide accurate information on how you process and fit data. The velocity of the structure is so far poorly resolved, especially at shallow depths. This can strongly affect the high frequency synthetic waveforms and thus your inference. How sensitive is the method to such velocity model uncertainties? You only show the fit for the "best" depth, but a reader has no idea what are the uncertainties... Could you plot the fit for perturbed depths as well? Next question is why only one station was used, since there are two of them at local distances. The analysis should be shown with both.

6. Minor comments:

L. 76: quantify "low frequencies"

Fig. 1: figure misses axis labels

Fig. 4: plots (or labels) should be enlarged, as labels are too small to be readable

Fig. 5: improve figure quality, it seems inadequate for the journal. There are no axes nor labels in plot c. If you add (too small) numbers in panel (a), they should refer to some events in the Figure or its caption.

Fig. 6 should show ALCN and ALCX

---

## Author Comment (AC1) · 9 Oct 2019

First of all we would like to thank the helpful and constructive comments made by both reviewers on the manuscript. While the comments are generally favorable, the reviewers raise a number of issues that we would like to respond in this rebuttal letter.

A common criticism of both reviewers is the size of the figures and their labels and symbols. While the original figures had a reasonable symbol and font size, when combining them into multi-panel figures, the size was reduced. To compensate for this deficiency we have redone most of the figures to increase the visibility of symbols and labels, and also to include some of the suggestions by the reviewers.

Moreover, in order to address some of the comments, and to facilitate the re-

producibility of the results presented here, we have created a data repository where all the data used and modeling results are available. The link to the repository (https://digital.csic.es/handle/10261/192082 ) and the DOI of the dataset (10.20350/digitalCSIC/8966) have been included in the "Data availability" and References sections of the manuscript.

Now we provide detailed responses to the reviewers comments (in italics), followed by our responses (in normal font).

Referee #1, Dr. Heather DeShon

General Comment: The manuscript "Fault reactivation by gas injection at an underground gas storage off the east coast of Spain" by A. Villasenor, R. Hermann, B. Gaite and A. Ugalde presents improved moment tensor solution for moderate magnitude earthquakes associated with induced earthquakes occurring offshore Spain in 2013. The motivation was to resolve a depth discrepancy for the earthquakes, which currently exists in the literature regarding the event sequence, in order to better understand the causal link between the gas storage facility, faults, and triggered seismicity. The study provides a careful analysis of moment tensors constrained using surface wave data and crustal reverberations to conclude that earthquake depths were between 6-9 km below the surface, in line with reactivation of presumed pre-existing NW-SE trending basement faults, rather than the 2 km depth in other papers consistent with injection levels. The study hypothesizes that to pore fluid pressure diffusion away from injection changes stress on the pre-existing fault structure enough to induce primarily strike-slip earthquakes consistent with the modern stress regime, in line with current research on induced earthquakes in Oklahoma, USA, for example. The paper requires minor changes to the text and figures to ensure consistency.

We are glad to see that we were able to convey the main objective of the manuscript, which is the discrepancy between the injection depth and the focal depths of the largest earthquakes of the sequence.
Specific Comments/Questions: The authors favor pore fluid pressure diffusion to link injection at <2 km to faulting at >6 km. The authors establish that there is a lack of geologic information for the crystalline basement in the study area publicly available. Is there any indication in the literature that faults in the basement offset the overlying units or that there is an extensive fault or fracture network that could serve to rapidly transmit fluid pressure?

The main faults in the region are extensional faults formed during the formation of the Valencia Trough. These faults are known to cut to the basement and could allow pore-fluid migration. We have added a reference about these faults.

Lines 384-385 hypothesize that faults in the basement have a different orientation than faults in the shallow geologic formations. Is there any evidence from the regional data that this could be the case?

We have also added a sentence with a reference about these basement faults.

Are there any faults that could be added to the figures to aid the reader in understanding the overall geologic setting? On lines 341-344 the authors reference faults as plotted in other studies but could these be added to the figures here for clarity?

According to the suggestion, we have plotted in Figure 1 the active faults included in the Quaternary Faults Database of Iberia (QAFI), which is the most complete and authoritative dataset of active faults in the region.

Is triggering via poroelastic stress change necessary?

Poro-elastic stress change is not the only mechanism for earthquake triggering. In fact a recent publication in Science (Bhattacharya and Viesca, 2019) suggests that aseismic fault slip could propagate faster and to larger distances that pore-fluid migration, which might be relevant for this case. Therefore we have added this reference and a small discussion to the manuscript.

Supplementary Material: It was not clear to me why the information in the supplement

(1 paragraph essentially and 1 figure) was not included in the main text. It seemed a valid question worth addressing in the main text. I leave it to the authors' decision however.

We agree with this suggestion. To incorporate it, we have eliminated panel b in Figure 4, and created a new Figure 5 with the comparison of goodness of fit versus focal depths for different frequency bands. However we have kept the Supplementary Material and increased it with another figure.

Citations: In addition to Yeck et al. and McNamara et al., this paper could cite a review paper such as Keranen and Wiengarten (2018), Induced Seismicity, Annual Review of Earth and Planetary Sciences, Vol. 46:149-174, https://doi.org/10.1146/annurev-earth-082517-010054

We agree with the suggestion and we have added this reference to the manuscript.

Figures: In general, the graphics clearly illustrate the points made in the main text. The fonts on the legends are very small, however. There is also a change in color scheme for data vs modeled waveforms in the main text and supplemental figure; red should be consistently used for modeled waveforms with blue/black used for observed data.

We have increased the size of the fonts in most figures, and used a consistent color scheme (red for data and blue for synthetics) in Figures 4, 7, and S1.

In Figure 2, the size of the circles make it difficult to tell the difference between EGF phase and group velocity (though of course the offset in c/U makes this clear).

We have made this figure in color to make it easier to distinguish the different symbols.

In Figure 5, the color bar is marked incorrectly. For example, red is 4 but having the 4 on the far left such that both 4 and 5 bound the red in the color box is not correct. This ends up making 9 and 10 km depth the same color, though there are at least 2 earthquakes at 10 km depth. Most importantly, what are the grey anastomosing lines? They are not referenced in the text or the caption for the figure.

To avoid confusion we have eliminated the triangles at the extremes of the color palette and added one more color for 10-11 km depth. This way earthquakes with different depths are represented with different colors.

The grey lines in Figures 5a,b represent the traces of faults at 1700 m depth obtained by Geostock (2010) from the more detailed 3D seismic studies carried out to delineate the reservoir size. This information should have been included, and we have fixed it in the revised version.

In Figure 6, the open circles and font sizes associated with the cross-correlation column are too small. The open circles can just be made solid, which may solve the small line width issue.

We have modified this figure (now Figure 7) to make labels and symbols clearly visible

Please also note the supplement to this comment:
https://www.solid-earth-discuss.net/se-2019-113/se-2019-113-AC1-supplement.pdf

[Figure]

**Supplement:**

**Response to reviewers of Solid Earth manuscript se-2019-113 "Fault reactivation by gas injection at an underground gas storage off the east coast of Spain" by A. Villaseñor et al.**

First of all we would like to thank the helpful and constructive comments made by both reviewers on the manuscript. While the comments are generally favorable, the reviewers raise a number of issues that we would like to respond in this rebuttal letter.

A common criticism of both reviewers is the size of the figures and their labels and symbols. While the original figures had a reasonable symbol and font size, when combining them into multi-panel figures, the size was reduced. To compensate for this deficiency we have redone most of the figures to increase the visibility of symbols and labels, and also to include some of the suggestions by the reviewers.

Moreover, in order to address some of the comments, and to facilitate the reproducibility of the results presented here, we have created a data repository where all the data used and modeling results are available. The link to the repository (https://digital.csic.es/handle/10261/192082) and the DOI of the dataset (10.20350/digitalCSIC/8966) have been included in the "Data availability" and References sections of the manuscript.

Now we provide detailed responses to the reviewers comments (in italics), followed by our responses (in normal font).

**Referee #1, Dr. Heather DeShon**

General Comment: The manuscript "Fault reactivation by gas injection at an underground gas storage off the east coast of Spain" by A. Villasenor, R. Hermann, B. Gaite and A. Ugalde presents improved moment tensor solution for moderate magnitude earthquakes associated with induced earthquakes occurring offshore Spain in 2013. The motivation was to resolve a depth discrepancy for the earthquakes, which currently exists in the literature regarding the event sequence, in order to better understand the causal link between the gas storage facility, faults, and triggered seismicity. The study provides a careful analysis of moment tensors constrained using surface wave data and crustal reverberations to conclude that earthquake depths were between 6-9 km below the surface, in line with reactivation of presumed pre-existing NW-SE trending basement faults, rather than the 2 km depth in other papers consistent with injection levels. The study hypothesizes that to pore fluid pressure diffusion away from injection changes stress on the pre-existing fault structure enough to induce primarily strike-slip earthquakes consistent with the modern stress regime, in line with current research on induced earthquakes in Oklahoma, USA, for example. The paper requires minor changes to the text and figures to ensure consistency.

We are glad to see that we were able to convey the main objective of the manuscript, which is the discrepancy between the injection depth and the focal depths of the largest earthquakes of the sequence.

Specific Comments/Questions: The authors favor pore fluid pressure diffusion to link injection at <2 km to faulting at >6 km. The authors establish that there is a lack of geologic information for the crystalline basement in the study area publicly available. Is there any indication in the literature that faults in the basement offset the overlying units or that there is an extensive fault or fracture network that could serve to rapidly transmit fluid pressure?

The main faults in the region are extensional faults formed during the formation of the Valencia Trough. These faults are known to cut to the basement and could allow pore-fluid migration. We have added a reference about these faults.

Lines 384-385 hypothesize that faults in the basement have a different orientation than faults in the shallow geologic formations. Is there any evidence from the regional data that this could be the case?

We have also added a sentence with a reference about these basement faults.

Are there any faults that could be added to the figures to aid the reader in understanding the overall geologic setting? On lines 341-344 the authors reference faults as plotted in other studies but could these be added to the figures here for clarity?

According to the suggestion, we have plotted in Figure 1 the active faults included in the Quaternary Faults Database of Iberia (QAFI), which is the most complete and authoritative dataset of active faults in the region.

**Is triggering via poroelastic stress change necessary?**

Poro-elastic stress change is not the only mechanism for earthquake triggering. In fact a recent publication in Science (Bhattacharya and Viesca, 2019) suggests that aseismic fault slip could propagate faster and to larger distances that pore-fluid migration, which might be relevant for this case. Therefore we have added this reference and a small discussion to the manuscript.

Supplementary Material: It was not clear to me why the information in the supplement (1 paragraph essentially and 1 figure) was not included in the main text. It seemed a valid question worth addressing in the main text. I leave it to the authors' decision however.

We agree with this suggestion. To incorporate it, we have eliminated panel b in Figure 4, and created a new Figure 5 with the comparison of goodness of fit versus focal depths for different frequency bands. However we have kept the Supplementary Material and increased it with another figure.

Citations: In addition to Yeck et al. and McNamara et al., this paper could cite a review paper such as Keranen and Wiengarten (2018), Induced Seismicity, Annual Review of

Earth and Planetary Sciences, Vol. 46:149-174, https://doi.org/10.1146/annurev-earth-082517-010054

We agree with the suggestion and we have added this reference to the manuscript.

Figures: In general, the graphics clearly illustrate the points made in the main text. The fonts on the legends are very small, however. There is also a change in color scheme for data vs modeled waveforms in the main text and supplemental figure; red should be consistently used for modeled waveforms with blue/black used for observed data.

We have increased the size of the fonts in most figures, and used a consistent color scheme (red for data and blue for synthetics) in Figures 4, 7, and S1.

In Figure 2, the size of the circles make it difficult to tell the difference between EGF phase and group velocity (though of course the offset in c/U makes this clear).

We have made this figure in color to make it easier to distinguish the different symbols.

In Figure 5, the color bar is marked incorrectly. For example, red is 4 but having the 4 on the far left such that both 4 and 5 bound the red in the color box is not correct. This ends up making 9 and 10 km depth the same color, though there are at least 2 earthquakes at 10 km depth. Most importantly, what are the grey anastomosing lines? They are not referenced in the text or the caption for the figure.

To avoid confusion we have eliminated the triangles at the extremes of the color palette and added one more color for 10-11 km depth. This way earthquakes with different depths are represented with different colors.

The grey lines in Figures 5a,b represent the traces of faults at 1700 m depth obtained by Geostock (2010) from the more detailed 3D seismic studies carried out to delineate the reservoir size. This information should have been included, and we have fixed it in the revised version.

In Figure 6, the open circles and font sizes associated with the cross-correlation column are too small. The open circles can just be made solid, which may solve the small line width issue.

We have modified this figure (now Figure 7) to make labels and symbols clearly visible.

**Anonymous referee #2**

The paper provides a seismological discussion on an interesting case of triggered seismicity in Europe, occurring in 2013 offshore Spain. The sequence was studied by a number of previous publications and reports. However, beside a general agreement on the relatively shallow hypocenters and strike-slip dominated mechanisms, accurate depth and fault geometry remain to a certain extent debated. Given the interest of the sequence and its relevant in the field of induced seismicity, this study appears to be justified.

Target of the study are basically on one side dispersion curves and velocity models, to improve Green's function and data modeling up to higher frequencies, and on the other side a contribution to the estimate of focal depths and focal mechanisms (or moment tensors).

Again we are glad to see that both reviewers understood the main message we wanted to convey with this manuscript.

*I think this is an interesting manuscript, but requires some moderate improvement. I provide below my major comments:*

Main comments:

**1. Uncertainties**

In order to provide new insights into a sequence which was discussed by previous papers, I think authors should not only provide a new result (depth, location or mechanism) but also some uncertainties. The estimation of uncertainties is discussed indeed in the first sections, dedicated to the assessment of dispersion curves and velocity models, but they are not used to derive a uncertainties on derived parameters, such as the depth.

In our study we have considered that it is more valuable to demonstrate that the main results and interpretations are supported by the data than to provide a rigorous error analysis, which is both difficult and not well established for a complex nonlinear problem such as moment tensor inversion.

For this reason we have made available in a repository all the data used, together with detailed modeling results. For all the 14 events analyzed (listed in Table 3) we provide in the repository the distribution of stations used, results of the grid search for focal depth, and waveform data fits. Resolution/uncertainty in focal depth can then be assessed by the reader by looking at plots like those shown in new Figure 5. When analyzing these plots we observe that in some cases the uncertainty in depth can be large (e.g. greater that 5 km), but it is also clear that for all events focal depths smaller than 4 km are not supported by the data. Since we do not interpret the actual value of focal depth, but the fact that it is significantly deeper than the injection depth, the evidence presented in the manuscript and in the repository supports our claims.

**2. Network asymmetry**

Both depth estimation, location and hypocenters suffer in this region by the asymmetric distribution of the stations. In this study, some new data have been taken into account (e.g. upon the Topoiberia project), but the azimuthal coverage remain strongly unbalanced. This may have a strong influence on the location accuracy, and some works suggested that the distribution plotted e.g. in Fig. 1b, may be partially attributed to the network geometry. The azimuthal coverage may also affect the depth,

**because of an inaccurate epicentral location. Has this been verified? Finally, it surely affects the focal mechanisms estimation. All these effects are not discusses.**

The location of the earthquakes in this sequence was addressed in a previous publication of our group (Gaite et al., 2016), and this is why it is not discussed in detail here. In that study we determined high precision traveltimes from waveform cross correlations, obtained a 3D velocity model for earthquake location, and located the earthquakes with a nonlinear method that produced realistic estimates of hypocentral uncertainties. After all this analysis the NW-SE orientation of the seismicity remained, so it is most likely a real feature and not only a result of the network geometry. This was discussed in lines 227-235 of the original manuscript.

**3. Data used for MT inversion**

Furthermore, authors use the same velocity model for all stations. While this may be proper for onshore stations, I doubt this is accurate for stations on Balearic islands. It is unclear whether these stations have been used or not, as they appear in Fig. 1 but not in Fig. 4. Using them will surely improve the coverage, and improve the moment tensor inversion result, but possibly a different velocity model should be used. Fig.4c should show some waveform fit there.

Figure 1 shows all permanent stations in the region, while Figure 4a shows the stations that were used for the earthquake analyzed in that figure. The repository contains a map for each earthquake showing the stations that were used to obtain that particular moment tensor. We have modified the captions of Figures 1 and 4 to make this clear.

Stations in the Balearic Islands were not used because of the reasons described by the reviewer. The 1D model used would not be appropriate for paths to those stations, and the inversion method used (Herrmann et al., 2011) does not perform well with marine/oceanic paths. Although we do have a 3D velocity model for the region (the one obtained by Gaite et al., 2016), it is computationally expensive to generate synthetic seismograms in 3D models, and we are not aware of any regional moment tensor inversion method that uses 3D models.

The effect of lack of station coverage to the east of the earthquakes is partly compensated by the fact that we fit the 3 components of the displacement: Z, R, and T Surface waves have the largest amplitudes at the distances and frequencies considered here, so using Z, R and T components means that we are fitting both Rayleigh and Love waves. Since Rayleigh and Love waves have different radiation patterns, even with an unfavorable station distribution, it is possible to obtain well determined nodal planes.

**4. Velocity models**

Since a lot of velocity models are discussed, they should be included in the document, as table or in the e-supplement. Having them available is need for the reproducibility of results.

The two models discussed in the manuscript are listed in Tables 1 and 2. This and other comments by reviewer 2 make us wonder whether he/she was not provided with a complete PDF or it was a low quality one (?). We agree with the reviewer that all the information necessary to reproduce the results should be available, and therefore we have created the aforementioned repository with all data used and modeling results.

**5. High frequency waveform modeling**

The high-frequency waveform comparison is very interesting and in my opinion the most interesting and novel part of the work. However, too little is said on how data were processed. Please, provide accurate information on how you process and fit data.

The only processing done to the data was to filter it (0.2-2 Hz band pass). This information has been added to section 5. Data fit is measured using the cross-correlation coefficient, and that is already specified in the manuscript.

The velocity of the structure is so far poorly resolved, especially at shallow depths. This can strongly affect the high frequency synthetic waveforms and thus your inference. How sensitive is the method to such velocity model uncertainties? You only show the fit for the "best" depth, but a reader has no idea what are the uncertainties... Could you plot the fit for perturbed depths as well

The shallow velocity structure in the area of the CASTOR UGS is not poorly known because there is a lot of information from well data. We have used this information to create our model for forward modeling of crustal reverberations (section 3.2), so the model uncertainties should be small. And the fact that we are able to match well the reverberations confirms that the model used is appropriate.

The value of the cross correlation coefficient as a function of depth shown in the right panels of Figure 7 provides information on the sensitivity to depth, particularly the low (poor fit) values obtained for depths lower than 4 km. However, showing the waveform comparison between synthetics and observed seismograms for different depths, as proposed by the reviewer, also provides very clear information about the sensitivity to depth. In fact we have produced these figures (such as Figure 9 in Frohlich et al., 2014) as intermediate results, but did no include them in the manuscript. To keep the balance between text and figures in the manuscript we have included one of those figures in the supplementary material. That figure clearly illustrates the poor fit to shallow depths, which is the main result of this manuscript.

**Next question is why only one station was used, since there are two of them at local distances. The analysis should be shown with both.**

The analysis was done with the closest station ALCN (25 km from CASTOR) because it was the only station in which the S wave reverberations were observed. The waveforms recorded at the second closest station, ALCX (40 km from CASTOR) did not exhibit clear reverberations (this could be caused by differences in seismic structure, attenuation of high frequencies, or other reasons) and therefore we did not try to match reverberations for that station.

**6. Minor comments: L. 76: quantify "low frequencies"**

We have modified the sentence to include the actual value of the frequency.

**Fig. 1: figure misses axis labels**

We do not know what the reviewer is exactly referring to. Both maps have latitude/longitude labels. In any case, we have increased the font size of the labels for better visibility.

**Fig. 4: plots (or labels) should be enlarged, as labels are too small to be readable**

To increase the visibility of the symbols and labels, and to address a comment by reviewer 1, this figure has been split in two, and the panels made larger.

Fig. 5: improve figure quality, it seems inadequate for the journal. There are no axes nor labels in plot c. If you add (too small) numbers in panel (a), they should refer to some events in the Figure or its caption.

We have increased the size of all panels and also have added axes with coordinates labeled in Figure 5c. Numbers in 5a above each beach ball corresponds to the entry of that event in Table 3. This is indicated in the caption, but since the caption is very long the reviewer might have overlooked it.

**Fig. 6 should show ALCN and ALCX**

We have previously explained why the crustal reverberation analysis was only done in ALCN.